# ADAPTIVE IMLE FOR FEW-SHOT IMAGE SYNTHESIS

## ABSTRACT

Despite their success on large datasets, GANs have been difficult to apply in the few-shot setting, where only a limited number of training examples are provided. Due to mode collapse, GANs tend to ignore some training examples, causing overfitting to a subset of the training dataset, which is small to begin with. A recent method called Implicit Maximum Likelihood Estimation (IMLE) is an alternative to GAN that tries to address this issue. It uses the same kind of generators as GANs but trains it with a different objective that encourages mode coverage. However, the theoretical guarantees of IMLE hold under restrictive conditions, such as the requirement for the optimal likelihood at all data points to be the same. In this paper, we present a more generalized formulation of IMLE which includes the original formulation as a special case, and we prove that the theoretical guarantees hold under weaker conditions. Using this generalized formulation, we further derive a new algorithm, which we dub Adaptive IMLE, which can adapt to the varying difficulty of different training examples. We demonstrate on multiple few-shot image synthesis datasets that our method significantly outperforms existing methods.

## 1 INTRODUCTION

Image synthesis has achieved significant progress over the past decade with the emergence of deep learning. Deep generative models such as GANs (Goodfellow et al., 2014; Brock et al., 2019; Karras et al., 2019; 2020; 2021), VAEs (Kingma & Welling, 2013; Vahdat & Kautz, 2020; Child, 2021; Razavi et al., 2019), diffusion models (Dhariwal & Nichol, 2021; Ho et al., 2020), score-based models (Song et al., 2021; Song & Ermon, 2019), normalizing flows (Dinh et al., 2017; Kobyzev et al., 2021; Kingma & Dhariwal, 2018), and autoregressive models (Salimans et al., 2017; van den Oord et al., 2016b;a) have made incredible improvements in generated image quality, which makes it possible to generate photorealistic images using these models.

Many of these deep generative models require training on a large-scale datasets to produce high-quality images. However, there are many real-life scenarios in that only a limited number of training examples are available, such as orphan diseases in the medical domain and rare events for training autonomous driving agents. One way to address this issue is by fine-tuning a model pre-trained on large auxiliary dataset from similar domains (Wang et al., 2020; Zhao et al., 2020a; Mo et al., 2020). Nonetheless, a large auxiliary dataset with a sufficient degree of similarity to the task at hand may not be available in all domains. If an insufficient similar auxiliary dataset were used regardless, image quality may be adversely impacted, as shown in (Zhao et al., 2020b). In this paper, we focus on the challenging setting of few-shot unconditional image synthesis without auxiliary pre-training.

The scarcity of training data in this setting makes it especially important for generative models to make full use of all training examples. This requirement sets it apart from the many-shot setting with abundant training data, where ignoring some training examples does not cause as big an issue. As a result, despite achieving impressive performance in the many-shot setting, GANs are challenging to apply to the few-shot setting due to the well-known problem of mode collapse, where the generator only learns from a subset of the training images and ignores the rest. A recent work (Li & Malik, 2018) proposed an alternative technique called Implicit Maximum Likelihood Estimation (IMLE) for unconditional image synthesis. Similar to GAN, IMLE uses a generator, but rather than adopting an adversarial objective which encourages each generated image to be similar to some training images, IMLE encourages each training image to have some similar generated images. Therefore, the generated images could cover all training examples without collapsing to a subset of the modes.

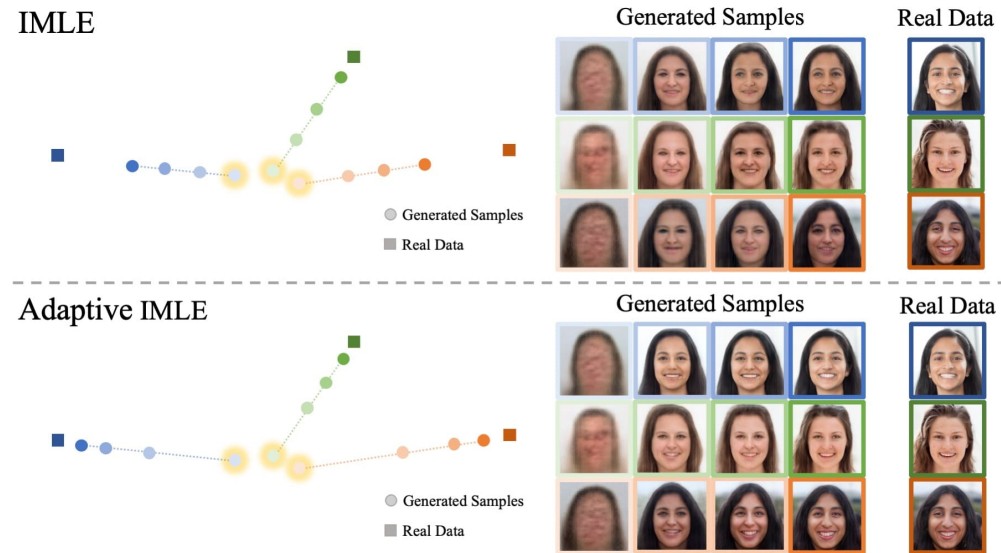

Figure 1: Schematic illustration that compares vanilla IMLE (Li & Malik, 2018) (top row) with the proposed algorithm, Adaptive IMLE (bottom row). While IMLE treats all training examples (denoted by the squares on the left) equally and pulls the generated samples (denoted by the circles on the left) towards them at a uniform pace, Adaptive IMLE adapts to the varying difficulty of each training example and pulls the generated samples towards them at an individualized pace that depends on the training example. The dashed line on the left figure illustrates the progression towards three data points at four comparable epochs with the starting positions highlighted. The corresponding generated samples are shown on the right. As shown, Adaptive IMLE can converge to the various data points faster and closer than IMLE.

However, the theoretical guarantees of IMLE hold under restrictive conditions, one of which is that all data points should have an identical optimal likelihood. The IMLE algorithm, therefore, treats all training examples equally when optimizing the model parameters and ignores the varying difficulty in learning from different training examples. As shown in the top row of Fig. 1, the generated samples make uneven progress toward different training examples using IMLE, leading to overfitting to some examples and underfitting to others. While this may not cause a major issue in the many-shot setting because many data points are expected to have similar optimal likelihoods, it can be quite problematic in the few-shot setting, since incorrectly weighting even a few training examples can impact the model quality substantially due to the small total number of training examples that the model is trained on.

In this paper, we introduce a generalized formulation of IMLE, which in turn enables the derivation of a new algorithm that requires fewer conditions and gets around the aforementioned issue. In particular, we mathematically prove that the theoretical guarantees of the generalized formulation hold under weaker conditions and subsumes the IMLE formulation as a special case. Furthermore, we derive an algorithm called *Adaptive IMLE* using this generalized formulation, which could adapt to points with different difficulties, as illustrated in the bottom row of Fig. 1. We compare our method to existing few-shot image synthesis baselines over six datasets and show significant improvements over the baselines in terms of mode modelling accuracy and coverage.

## 2 RELATED WORK

There are two broad families of work on few-shot learning, one that focuses on discriminative tasks such as classification (O' Mahony et al., 2019; Finn et al., 2017; Snell et al., 2017) and another that focuses on generative tasks. In this paper, we focus on the latter. Similar to many-shot generation tasks, few-shot generation tasks take a limited number of training examples as input and aim to generate samples that are similar to those training examples. What is different from the many-shot

setting is that it is crucial for the generative model to utilize all the training examples in the few-shot setting. Due to the scarcity of available data for training, ignoring even just a few data points would cause a more serious issue in the few-shot setting than in the many-shot setting. One line of work focuses on pre-training on large-scale auxiliary datasets from similar domains and adapting the pre-trained models for the few-shot task. This has been applied to unconditional image generation (Li et al., 2020; Zhao et al., 2020a; Mo et al., 2020; Ojha et al., 2021; Wang et al., 2020), conditional image generation (Sinha et al., 2021; Liu et al., 2019) and video generation (Wang et al., 2019). However, there are no guarantees on the existence of such large-scale auxiliary datasets for all domains, and recent studies (Zhao et al., 2020b; Kong et al., 2022) also showed that fine-tuning from a dissimilar domain could even lead to the degradation of generated image quality.

In this paper, we focus on the setting without fine-tuning pre-trained models from auxiliary datasets. Most prior work considered applying GANs to this setting and developed methods for alleviating the well-known mode collapse problem of GANs. FastGAN (Liu et al., 2021) introduced a skip-layer excitation module for faster training and used self-supervision for the discriminator to learn more descriptive features, which aids better mode coverage of the generator. MixDL (Kong et al., 2022) introduced a two-sided distance regularization to facilitate learning smooth and mode-preserving latent space. Despite these improvements, some degree of mode collapse still remains. A recent method called Implicit Maximum Likelihood Estimation (IMLE) (Li & Malik, 2018) adopted a different objective function and showed promising results towards alleviating mode collapse on unconditional image synthesis tasks. Prior IMLE-based methods mainly focused on conditional image synthesis (Li* et al., 2020; Peng et al., 2022). In this work, we build on (Li & Malik, 2018) and introduce a novel and more generalized formulation of IMLE to make it more suitable for the unconditional few-shot setting.

## 3 BACKGROUND: IMPLICIT MAXIMUM LIKELIHOOD ESTIMATION (IMLE)

In unconditional image synthesis, the goal is to learn the unconditional probability distribution of images $p(\mathbf{x})$, from which samples can be drawn to yield new synthesized images. It is common to use an implicit generative model – one example of such a model is the generator in GANs, which takes the form of a function $T_\theta$ parameterized as a neural network with parameters $\theta$, which maps latent codes $\mathbf{z}$ drawn from a standard Gaussian $\mathcal{N}(0, \mathbf{I})$ to images $\mathbf{x}$. One way to learn this model is with the GAN objective, which introduces a discriminator that aims to distinguish between generated images $T_\theta(\mathbf{z})$ and real images $\mathbf{x}$. The generator is trained to produce more realistic images that would fool the discriminator. However, the output $T_\theta(\mathbf{z})$ tends to recover only a subset of the training examples even when varying all values of $\mathbf{z}$. This issue is known as mode collapse, and the intuitive reason behind it is that the adversarial objective of GAN only encourages each generated sample to be similar to some training examples, but there is no guarantee that all training examples will have some similar generated samples. In the few-shot image synthesis setting, the issue of mode collapse is even more significant given the limited number of training examples that are available in the first place.

A more recent method known as Implicit Maximum Likelihood Estimation (IMLE) (Li & Malik, 2018) proposed an alternative objective to address this issue. Instead of making each generated sample similar to *some* training examples, IMLE tries to ensure that samples can be generated around *each* training example $\mathbf{x}_i$. The generator $T_\theta$ is encouraged to pull some samples $T_\theta(\mathbf{z}_j)$ towards each $\mathbf{x}_i$, thereby rewarding coverage of the modes associated with all training examples.

More precisely, the IMLE objective takes the following form:

$$\min_\theta \mathbb{E}_{z_1,\ldots,z_m \sim \mathcal{N}(0,I)} \left[ \sum_{i=1}^{n} \min_{j \in [m]} d\left(\mathbf{x}_i, T_\theta(\mathbf{z}_j)\right) \right] \tag{1}$$

where $d(.,.)$ is a distance metric, $m$ is a hyperparameter, and $\mathbf{x}_i$ is the $i^{th}$ training example. The training procedure involves finding the nearest generated sample index $\sigma(i)$ to each training example $\mathbf{x}_i$, and optimizing the model parameter $\theta$ by minimizing the distance from the selected sample $T_\theta\left(\mathbf{z}_{\sigma(\mathbf{i})}\right)$ to the target data $\mathbf{x}_i$. Detailed pseudocode of the algorithm can be found in the appendix.

Despite the algorithm's simplicity, restrictive conditions need to be satisfied for the theoretical guarantees of IMLE to hold, such as requiring a uniform optimal likelihood for all data points. As an example, consider a dataset with two clusters with the same number of points where one cluster

has large variance and the other has small variance. In this case, the training examples from the high-variance cluster are more difficult to learn than the training examples from the low-variance cluster, because of sparser coverage of the space in the former cluster. If we consider what the ground truth data distribution looks like, it is a bimodal distribution, with the mode corresponding to the low-variance cluster having higher likelihood than the other. So requiring uniform optimal likelihood for all data points, as IMLE does, will result in overfitting to the low-variance cluster and underfitting to the high-variance cluster, which is not optimal. We refer readers to the IMLE paper (Li & Malik, 2018) for more details.

# 4 METHOD

In this paper, we devise a generalized formulation of IMLE, whose theoretical guarantees hold under more general conditions than vanilla IMLE (Sec 4.1). This formulation subsumes vanilla IMLE as a special case and also gives rise to a new algorithm which we call *Adaptive IMLE*. It turns out that Adaptive IMLE offers theoretical and practical advantages over IMLE, which we will demonstrate (Sec 4.2).

## 4.1 GENERALIZED FORMULATION

Since $T_\theta$ is an implicit generative model, the likelihood induced by the model $p_\theta$ cannot in general be expressed in closed form, and so evaluating it numerically is typically computationally intractable. In order to train the generative model, we would like to maximize the likelihood of the training examples without actually needing to evaluate the likelihood. Below we will consider the generalized objective we propose and show that optimizing the objective is equivalent to maximizing the sum of likelihoods at the training examples, without requiring the evaluation of likelihood.

Consider the following optimization problem:

$$\max_\theta \mathcal{L}_{\{\tau_i\}_i}(\theta) := \max_\theta \mathbb{E}_{z_1,\ldots,z_m \sim \mathcal{N}(0,I)} \left[ \frac{1}{n} \sum_{i=1}^n \frac{1}{w_i} \left( \tau_i - \frac{1}{m} \sum_{j=1}^m \Phi_{\tau_i}(d(\mathbf{x}_i, T_\theta(\mathbf{z}_j))) \right) \right] \quad (2)$$

where $T_\theta$, $d(\cdot, \cdot)$ and $m$ are as defined in Eqn 1. We will choose $w$, $\tau$ and $\Phi_\tau(\cdot)$ later based on the insight revealed by lemmas below.

We will present the high-level sketches of our key lemmas (omitting some technicalities) and delineate their interpretations and significance. The precise statements of the lemmas and their proofs are left to the appendix.

We will first present a lemma that relates an expectation of a random variable to the weighted integral of one minus its cumulative density function (CDF) evaluated at different points, which we will refer to as cumulative densities.

**Lemma 1.** *Let $X$ be a non-negative random variable and $\Phi$ be a continuous function on $[0, \infty)$. If $\Phi'$ is integrable on all closed intervals in $[0, \infty)$,*

$$\mathbb{E}[\Phi(X)] = \Phi(0) + \int_0^\infty \Phi'(t) \Pr(X \geq t) dt$$

This lemma is useful because the left-hand side (LHS) is easy to approximate with Monte Carlo estimates of expectations, and the right-hand side (RHS) is a weighted integral of one minus cumulative densities, which are intractable to compute in general. It enables us to control the weighting of different cumulative densities by choosing the function $\Phi$.

Recall that our goal is to maximize the likelihood at each training example without actually computing the likelihood. We can leverage Lemma 1 for this purpose, by choosing the non-negative random variable $X$ appropriately. We choose $X$ to be the distance between a training example and a generated sample $d(\mathbf{x}_i, T_\theta(\mathbf{z}_j))$. With this choice, Lemma 1 gives us a way to relate a weighted integral of the average likelihoods within differently sized neighbourhoods around the training example $\mathbf{x}_i$ (RHS) to the expectation of a function of the distance $d(\mathbf{x}_i, T_\theta(\mathbf{z}_j))$ (LHS).

Moreover, we'd like to restrict the average likelihoods we integrate over to only those within neighbourhoods of certain sizes rather than from 0 to $\infty$. Specifically, we'd like to integrate from $\delta\tau$ to $\tau$, where $\tau > 0$ is the radius of the largest neighbourhood and $0 \leq \delta < 1$ is a tightening threshold. To this end, we can choose the weighting function $\Phi'_\tau(\cdot)$ to be 1 when $\delta\tau \leq t \leq \tau$ and 0 otherwise. One choice of such $\Phi_\tau(\cdot)$ that satisfies this condition and its associated $\Phi'_\tau(\cdot)$ are:

$$\Phi_\tau(t) = \begin{cases} \delta\tau & t < \delta\tau \\ t & \delta\tau \leq t \leq \tau \\ \tau & t > \tau \end{cases} \qquad \Phi'_\tau(t) = \begin{cases} 0 & t < \delta\tau \\ 1 & \delta\tau \leq t \leq \tau \\ 0 & t > \tau \end{cases}$$

Using this choice of $\Phi_\tau(\cdot)$, we obtain the following lemma for a particular training example $\mathbf{x}_i$.

**Lemma 2.** *Under the choice of $\Phi_\tau(\cdot)$ above and its associated $\Phi'_\tau(\cdot)$,*

$$\mathbb{E}_{z_1,\ldots,z_m \sim \mathcal{N}(0,I)} \left[\Phi_{\tau_i}(d(\mathbf{x}_i, T_\theta(\mathbf{z}_j)))\right] = \tau_i - \int_{\delta\tau_i}^{\tau_i} \Pr(d(\mathbf{x}_i, T_\theta(\mathbf{z}_j)) < t)dt.$$

This lemma shows that, for one training example $\mathbf{x}_i$, the expectation on the LHS reduces to $\tau_i$ minus the integral of the average likelihoods within balls whose radii lie between $\delta\tau_i$ and $\tau_i$. Applying Lemma 2 to all training examples $\mathbf{x}_1, \ldots, \mathbf{x}_n$, we obtain the following lemma that reveals what the overall objective in Eqn 2 optimizes.

**Lemma 3.** *Under the choice of $\Phi_\tau(\cdot)$ above and its associated $\Phi'_\tau(\cdot)$,*

$$\mathcal{L}_{\{\tau_i\}_i}(\theta) = \frac{1}{n} \sum_{i=1}^{n} \frac{1}{mw_i} \sum_{j=1}^{m} \int_{\delta\tau_i}^{\tau_i} \Pr(d(\mathbf{x}_i, T_\theta(\mathbf{z}_j)) < t)dt.$$

Lemma 3 shows that $\mathcal{L}_{\{\tau_i\}_i}(\theta)$ implicitly computes the average likelihood that the generative model assigns to the neighbourhood of each data point. The choice of $\tau_i$ controls the radius. Since we would like to maximize probability in the immediate neighbourhood of each data point, we would like $\tau_i$ to be small.

So should we choose an arbitrarily small value for $\tau_i$? Recall that by definition of $\Phi_{\tau_i}(\cdot)$, if $d(\mathbf{x}_i, T_\theta(\mathbf{z}_j)) > \tau_i$, $\Phi_{\tau_i}(d(\mathbf{x}_i, T_\theta(\mathbf{z}_j))) = \tau_i$. So, for a very small $\tau_i$, it may well be the case that $d(\mathbf{x}_i, T_\theta(\mathbf{z}_j)) > \tau_i \; \forall j$, which would make the Monte Carlo estimate of $\mathcal{L}_{\{\tau_i\}_i}(\theta)$, i.e., $\frac{1}{n} \sum_{i=1}^{n} \frac{1}{w_i} \left(\tau_i - \frac{1}{m} \sum_{j=1}^{m} \Phi_{\tau_i}(d(\mathbf{x}_i, T_\theta(\mathbf{z}_j)))\right)$, zero. Since this is a constant, the gradient w.r.t. the parameters is zero, which makes gradient-based learning impossible. This would happen whenever $\tau_i < \min_{j \in [m]} d(\mathbf{x}_i, T_\theta(\mathbf{z}_j))$, and so the smallest $\tau_i$ that can be chosen is $\min_{j \in [m]} d(\mathbf{x}_i, T_\theta(\mathbf{z}_j))$ (which is treated as a constant rather than a function of $\theta$).

With this choice of $\tau_i$, assuming that there is a unique $j^*$ such that $d(\mathbf{x}_i, T_\theta(\mathbf{z}_{j^*})) = \min_{j \in [m]} d(\mathbf{x}_i, T_\theta(\mathbf{z}_j))$ (which happens almost surely), the objective can be simplified to:

$$\mathcal{L}_{\{\tau_i\}_i}(\theta) = \mathbb{E}_{z_1,\ldots,z_m \sim \mathcal{N}(0,I)} \left[\frac{1}{nm} \sum_{i=1}^{n} \frac{1}{w_i} \left(\tau_i - \max(\min_{j \in [m]} d(\mathbf{x}_i, T_\theta(\mathbf{z}_j)), \delta\tau_i)\right)\right] \qquad (3)$$

If we minimize the objective in Eqn. 3, we get a novel objective known as the Adaptive IMLE objective. The solution to the Adaptive IMLE objective can be expressed as:

$$\arg\max_\theta \mathcal{L}_{\{\tau_i\}_i}(\theta) = \arg\min_\theta \mathbb{E}_{z_1,\ldots,z_m \sim \mathcal{N}(0,I)} \left[\sum_{i=1}^{n} \frac{1}{w_i} \max(\min_{j \in [m]} d(\mathbf{x}_i, T_\theta(\mathbf{z}_j)), \delta\tau_i)\right] \qquad (4)$$

It turns out that the vanilla IMLE objective can be recovered as a special case, by choosing $\delta = 0$ and $w_1 = w_2 = \cdots = w_n$.

$$
\begin{aligned}
\arg\max_\theta \mathcal{L}_{\{\tau_i\}_i}(\theta) &= \arg\min_\theta \mathbb{E}_{z_1,\ldots,z_m \sim \mathcal{N}(0,I)} \left[ \sum_{i=1}^{n} \frac{1}{w_i} \max(\min_{j\in[m]} d\left(\mathbf{x}_i, T_\theta(\mathbf{z}_j)\right), 0) \right] \\
&= \arg\min_\theta \mathbb{E}_{z_1,\ldots,z_m \sim \mathcal{N}(0,I)} \left[ \sum_{i=1}^{n} \frac{1}{w_i} \min_{j\in[m]} d\left(\mathbf{x}_i, T_\theta(\mathbf{z}_j)\right) \right] \\
&= \arg\min_\theta \mathbb{E}_{z_1,\ldots,z_m \sim \mathcal{N}(0,I)} \left[ \sum_{i=1}^{n} \min_{j\in[m]} d\left(\mathbf{x}_i, T_\theta(\mathbf{z}_j)\right) \right]
\end{aligned}
$$

### 4.1.1 CURRICULUM LEARNING

Recall that our goal is to maximize the likelihood of the immediate neighbourhood around each data point, and the size of this neighbourhood is controlled by $\tau_i$. Therefore, we want to make $\tau_i$ small. In order to make $\tau_i$ small without impeding learning, we need to make $\mathbb{E}_{z_1,\ldots,z_m \sim \mathcal{N}(0,I)}[\tau_i] = \mathbb{E}_{z_1,\ldots,z_m \sim \mathcal{N}(0,I)} \left[\min_{j\in[m]} d(\mathbf{x}_i, T_\theta(\mathbf{z}_j))\right]$ small. To this end, we can either increase $m$, the number of samples, or train $T_\theta$ so that the samples it produces are close to the data point $\mathbf{x}_i$. The former is computationally expensive, and so we will devise a method to achieve the latter.

We propose a curriculum learning strategy, which solves a sequence of optimization problems with different $\tau_i$'s, such that $\tau_i$'s get smaller for optimization problems later in the sequence. The earlier optimization problems in the sequence help train $T_\theta$ to produce samples close to the data points. After each optimization problem is solved to convergence, we start solving the next optimization problem with $\theta$ initialized to the solution found previously.

This will make $\tau_i$'s smaller and smaller. If they eventually converge to zero, then it turns out that we would have equivalently maximized the sum of likelihoods $p_\theta(\mathbf{x}_i)$ of the training examples under the probability distribution induced the generative model, as shown in the lemma below.

**Lemma 4.** *Suppose $p_\theta$ is continuous at all data points $\mathbf{x}_1, \ldots, \mathbf{x}_n$, under the choice of $w_i = \int_{\delta\tau_i}^{\tau_i} \mathrm{vol}(B_t(\mathbf{x}_i))dt := \int_{\delta\tau_i}^{\tau_i} \int_{B_t(\mathbf{x}_i)} d\mathbf{x}dt$, where $B_r(\mathbf{x}) = \{\mathbf{y}|d(\mathbf{y},\mathbf{x}) < r\}$ is an open ball of radius $r$ centred at $\mathbf{x}$,*

$$
\lim_{\{\tau_i \to 0^+\}_i} \mathcal{L}_{\{\tau_i\}_i}(\theta) = \frac{1}{n}\sum_{i=1}^{n} p_\theta(\mathbf{x}_i)
$$

This lemma shows the theoretical guarantees of Adaptive IMLE hold under more general conditions that those required by vanilla IMLE.

### 4.2 ADAPTIVE IMLE

The key difference from the objective in Eqn. 4 to the original IMLE formulation in Eqn. 1 is the individualized neighbourhood radius $\tau_i$ around each data point $\mathbf{x}_i$. This change in the objective is crucial, as it allows the model to adapt to the varying difficulty in learning different training examples, hence the algorithm name, *Adaptive IMLE*.

As mentioned in Sect. 4.1.1, we need to gradually decrease $\tau_i$ in order to make the learning feasible. This could be achieved by decreasing the tightening threshold $\delta\tau_i$. To this end, the algorithm optimizes the model's parameter until the distance between the generated sample and the target data $d(\mathbf{x}_i, T_\theta(\mathbf{z}_j))$ decreases to $\delta\tau_i$. Once the threshold is reached, the algorithm decreases the threshold by multiplying it by $\delta$ as $0 \le \delta < 1$. This updated threshold then serves as the new target for learning $\mathbf{x}_i$. Intuitively, the tightening coefficient $\delta$ determines the amount of progress required for the selected sample towards each training example.

Now let's turn our attention to the optimization problem we solve in each stage of the curriculum. Consider the following unweighted variant to the objective in Eqn. 4 without the $\frac{1}{w_i}$ factor:

$$
\arg\min_\theta \mathbb{E}_{z_1,\ldots,z_m \sim \mathcal{N}(0,I)} \left[ \sum_{i=1}^{n} \max(\min_{j\in[m]} d(\mathbf{x}_i, T_\theta(\mathbf{z}_j)), \delta\tau_i) \right] \tag{5}
$$

---

**Algorithm 1** Adaptive IMLE Procedure

---

**Require:** The set of inputs $\{\mathbf{x}_i\}_{i=1}^n$, tightening coefficient $\delta \in [0, 1)$
1: Initialize the parameters $\theta$ of the generator $T_\theta$
2: Draw latent codes $Z \leftarrow \mathbf{z}_1, ..., \mathbf{z}_m$ from $\mathcal{N}(0, \mathbf{I})$
3: $\sigma(i) \leftarrow \arg\min_{j \in [m]} d(\mathbf{x}_i, T_\theta(\mathbf{z}_j)) \ \forall i \in [n]$
4: $\tau_i \leftarrow d\left(\mathbf{x}_i, T_\theta\left(\mathbf{z}_{\sigma(i)}\right)\right) \ \forall i \in [n]$               ▷ Initialize the threshold for each data point
5: **for** $k = 1$ **to** $K$ **do**
6:      Pick a random batch $S \subseteq [n]$
7:      $\theta \leftarrow \theta - \eta \nabla_\theta \left(\sum_{i \in S} d\left(\mathbf{x}_i, T_\theta\left(\mathbf{z}_{\sigma(i)}\right)\right)\right)/|S|$
8:      Draw latent codes $Z \leftarrow \mathbf{z}_1, ..., \mathbf{z}_m$ from $\mathcal{N}(0, \mathbf{I})$
9:      **for** $i \in S$ **do**
10:         **if** $d\left(\mathbf{x}_i, T_\theta\left(\mathbf{z}_{\sigma(i)}\right)\right) \leq \delta\tau_i$ **then**      ▷ Only update $\sigma(i)$ when getting into the threshold
11:            $\tau_i \leftarrow \tau_i \delta$                                    ▷ Tightening the threshold
12:            $\sigma(i) \leftarrow \arg\min_{j \in [m]} d(\mathbf{x}_i, T_\theta(\mathbf{z}_j))$
13:         **end if**
14:      **end for**
15: **end for**
16: **return** $\theta$

---

If we were to run stochastic gradient descent (SGD) without replacement on the weighted objective (Eqn. 4) and the unweighted objective (Eqn. 5) and consider the updates made by each, we will find that updates induced by the weighted objective are just scalar multiples of those induced by the unweighted objective, since $\tau_i$ and therefore $w_i$ is fixed during each stage of the curriculum. So, optimizing the weighted objective is equivalent to optimizing the unweighted objective, with a different step size chosen for each update. We consider optimizing the unweighted objective with a constant step size that is smaller than these per-iteration step sizes. Then, if SGD on the weighted objective converges to a solution [1] where $\min_{j \in [m]} d(\mathbf{x}_i, T_\theta(\mathbf{z}_j))$ falls below $\delta\tau_i$ (which is a global minimum), SGD on the unweighted objective with such a step size will do the same, because the latter is just choosing more conservative step sizes than implied by the weighted objective.

Now, if we putting everything together, we obtain the Adaptive IMLE algorithm. The details are shown in Algorithm 1.

## 5 EXPERIMENTS

**Baselines** We compare our method to recent few-shot unconditional image synthesis methods that operate in the same setting we consider, namely without needing to pre-train on auxiliary datasets. Two of such recent methods are FastGAN (Liu et al., 2021) and MixDL (Kong et al., 2022).

**Training Details** Our network architecture is modified from Child (2021), where we keep the decoder architecture and replace the encoder with a fully-connected mapping network inspired by Karras et al. (2019). We choose an input latent dimension of $1024$ and a tightening coefficient $\delta = 0.9$. We train our model for 500k iterations with a mini-batch size of 4 using the Adam optimizer (Kingma & Ba, 2015) with a learning rate of $2 \times 10^{-6}$ on a single NVIDIA V100 GPU.

**Datasets** We evaluate our method and the baselines on a wide range of natural image datasets at $256 \times 256$ resolution, which includes Animal-Face Dog and Cat (Si & Zhu, 2012), Obama, Panda, and Grumpy-cat (Zhao et al., 2020b) and Flickr-FaceHQ (FFHQ) subset (Karras et al., 2019). All datasets contain 100 images except for Dog and Cat which contain 389 and 160 images respectively. The FFHQ subset consists of 100 FFHQ images with similar backgrounds, in order to highlight diversity in the generation of foregrounds.

**Evaluation Metrics** We use the Fréchet Inception Distance (FID) (Heusel et al., 2017) to measure the perceptual quality of the generated images, where we randomly generate 5000 images and compute FID between the generated samples and real images in each dataset. To evaluate the mode modelling accuracy (precision) and coverage (recall), we use the precision metric of Kynkäänniemi et al. (2019) to measure the former, and use the recall metric of Kynkäänniemi et al. (2019) and

---

[1]The weighted objective will always converge to such a solution since we can choose $\delta$ to be close to 1.

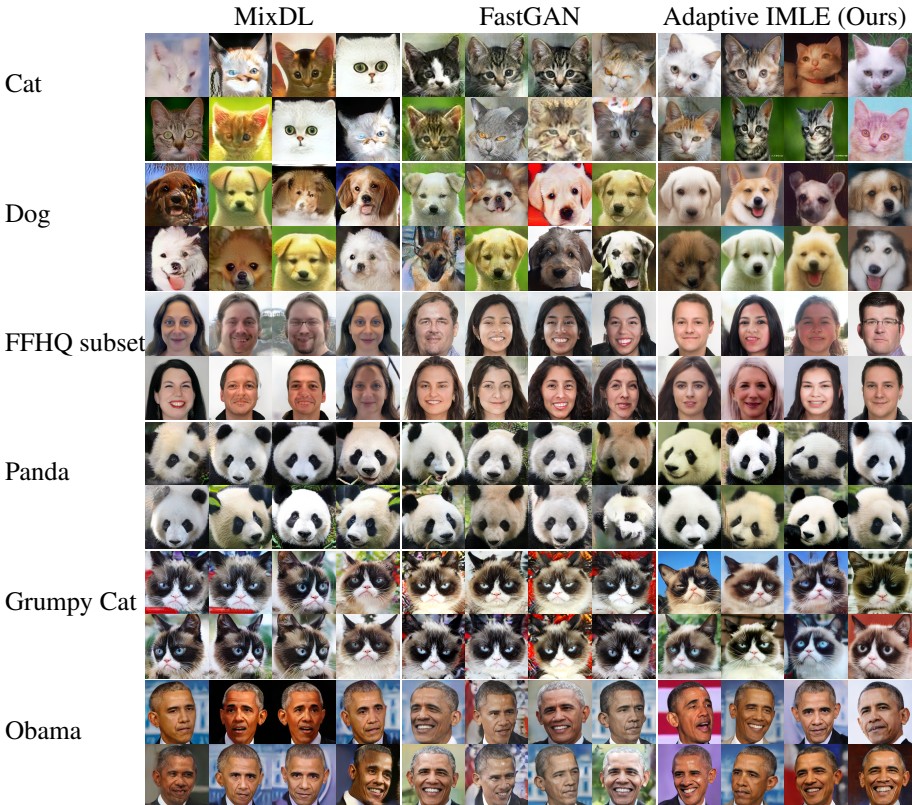

Figure 2: Qualitative comparison of images generated by our method and those generated by the baselines, FastGAN (Liu et al., 2021) and MixDL (Kong et al., 2022). As shown, our samples are of higher quality and have greater diversity. On the other hand, the samples generated by the baselines show more limited diversity, which is validated by the recall results in Table. 2, which suggest that the baselines exhibit mode collapse.

| | Grumpy Cat | | Obama | | Panda | | Cat | | Dog | | FFHQ subset | |
|---|---|---|---|---|---|---|---|---|---|---|---|---|
| | FID↓ | LPIPS↓ | FID↓ | LPIPS↓ | FID↓ | LPIPS↓ | FID↓ | LPIPS↓ | FID↓ | LPIPS↓ | FID↓ | LPIPS↓ |
| *FastGAN (Liu et al., 2021)* | 26.6 | 0.357 | 41.1 | 0.370 | 10.0 | 0.339 | 35.1 | 0.467 | 50.7 | 0.430 | 54.2 | 0.357 |
| *MixDL(Kong et al., 2022)* | 24.5 | 0.296 | 45.4 | 0.276 | 10.6 | 0.264 | **26.5** | 0.305 | 81.2 | 0.274 | 62.3 | 0.221 |
| *Adaptive IMLE (Ours)* | **23.4** | **0.058** | **30.7** | **0.036** | **8.5** | **0.039** | 29.9 | **0.074** | **50.6** | **0.072** | **43.9** | **0.014** |

Table 1: We compute FID (Heusel et al., 2017) between the real data and 5000 randomly generated samples in all cases. LPIPS above represents LPIPS backtracking score (Liu et al., 2021). For this metric, each model is trained on 90% of the dataset. The resulting model is used to backtrack in the latent space and reconstruct the remaining 10%. Lower LPIPS backtracking score shows better mode coverage of the training data.

LPIPS backtracking score (Liu et al., 2021) to measure the latter. For LPIPS backtracking, we use 90% of the full dataset for training and evaluate the metric using the remaining 10% of the dataset.

## 5.1 QUANTITATIVE RESULTS

We compare the FID and LPIPS backtracking scores across all methods in Tab. 1. As shown, our method outperforms the baselines in terms of both metrics on all datasets except for Cat, where our FID is the second best and LPIPS backtracking score is the best. We compare the mode accuracy and coverage in Tab. 2. As shown, our method achieves better precision than the baselines and significantly outperforms the baselines in terms of recall. These results show that our method could produce high-quality images while obtaining better mode coverage compared to the baselines.

|  | Grumpy Cat | | Obama | | Panda | | Cat | | Dog | | FFHQ subset | |
|---|---|---|---|---|---|---|---|---|---|---|---|---|
|  | Prec.↑ | Rec.↑ | Prec.↑ | Rec.↑ | Prec.↑ | Rec.↑ | Prec.↑ | Rec.↑ | Prec.↑ | Rec.↑ | Prec.↑ | Rec.↑ |
| *FastGAN (Liu et al., 2021)* | 0.909 | 0.13 | 0.918 | 0.09 | 0.957 | 0.16 | 0.971 | 0.08 | 0.961 | 0.19 | 0.913 | 0.13 |
| *MixDL(Kong et al., 2022)* | 0.931 | 0.35 | 0.910 | 0.47 | 0.933 | 0.30 | 0.910 | 0.50 | 0.857 | 0.15 | 0.770 | 0.30 |
| *Adaptive IMLE (Ours)* | **0.987** | **0.73** | **0.967** | **0.85** | **0.990** | **0.87** | **0.972** | **0.71** | **0.972** | **0.48** | **0.998** | **0.76** |

Table 2: Precision and recall (Kynkäänniemi et al., 2019) is computed across 1000 randomly generated samples and the target dataset. Our method performs better for both precision and recall in all cases. Higher precision shows better fitting to the target dataset and higher recall corresponds to better mode coverage.

## 5.2 QUALITATIVE RESULTS

We show the qualitative comparison of our method to the baselines in Fig. 2. As shown, our method generates higher quality samples which better preserve the semantic structures compared to the baselines, such as the eyes in Cat, the facial structure in Dog and the mouth and hair in the FFHQ subset. In addition, our method generates more diverse results while the baselines suffer from mode collapse and generate similar samples, such as in Panda, Grumpy Cat and Obama. Additional samples from our model can be found in the appendix.

We show the final reconstruction of the target image found using LPIPS backtracking (Liu et al., 2021) on the models trained with different methods in Fig. 3. As shown, our method is the only one where the reconstruction is structurally similar to the target image, demonstrating that our model successfully covers the mode that the target image belongs to.

We also compare our method to the baselines on the quality of interpolations between two samples in the latent space. As shown in Fig. 4, our method interpolates more smoothly and naturally than the baselines, thereby indicating that our model is less overfitted to the training examples.

Target image MixDL FastGAN Ours Target image MixDL FastGAN Ours

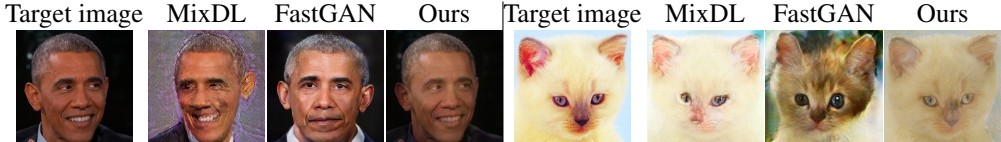

Figure 3: Visualizations of the reconstructions of an unseen target image from LPIPS backtracking. While the reconstructions of MixDL and FastGAN are structurally dissimilar from the target images, the reconstructions of our method are structurally similar to the target images.

         Obama             FFHQ subset

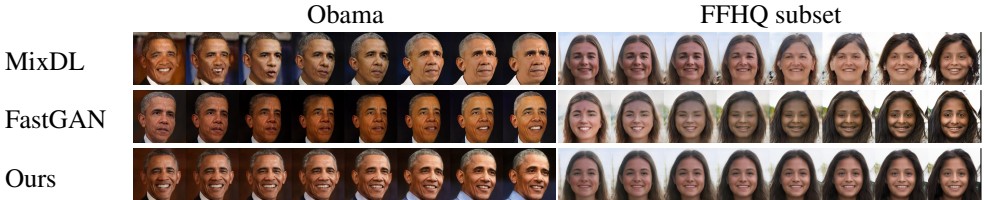

Figure 4: Latent space interpolation results. Our results show smooth and meaningful transitions and higher quality images generated from intermediate points along the interpolation line in the latent space. Start and end of interpolations are nearest neighbours of the same data examples among 200 samples generated by each method.

## 5.3 ABLATION STUDY

We compare the FID, precision, and recall between the proposed method, Adaptive IMLE, and vanilla IMLE on the more challenging datasets, Obama and FFHQ subset. As shown in Tab. 3, Adaptive IMLE significantly improves upon vanilla IMLE in terms of FID and recall while achieving similar precision, validating the effectiveness of the proposed method under the few-shot setting.

| | Obama | | | FFHQ subset | | |
|---|---|---|---|---|---|---|
| | FID↓ | Prec.↑ | Rec.↑ | FID↓ | Prec.↑ | Rec.↑ |
| *Vanilla IMLE (Li & Malik, 2018)* | 37.4 | **0.973** | 0.61 | 54.1 | **0.999** | 0.51 |
| *Adaptive IMLE (Ours)* | **30.7** | 0.967 | **0.85** | **43.9** | 0.998 | **0.76** |

Table 3: Adaptive IMLE significantly improves perceptual quality (FID) and recall compared to vanilla IMLE, while maintaining similarly high levels of precision.

## 6 CONCLUSION

We developed a method for the challenging few-shot image synthesis setting that does not depend on pre-training on auxiliary datasets. We presented a more generalized formulation of IMLE and proved that the theoretical guarantees of this generalized formulation hold under weaker conditions. We further derived a novel algorithm based on this formulation which can adapt to different training examples of varying difficulty. We showed that our method significantly outperforms existing baselines in terms of mode modelling accuracy and coverage on six few-shot benchmark datasets.

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

## A    PROOFS

**Lemma 1.** *Let $X$ be a non-negative random variable and $\Phi$ be a continuous function on $[0, \infty)$. If $\Phi'$ is integrable on all closed intervals in $[0, \infty)$,*

$$\mathbb{E}\left[\Phi(X)\right] = \Phi(0) + \int_0^\infty \Phi'(t)\Pr(X \geq t)dt$$

*Proof.*

$$
\begin{aligned}
\Phi(0) + \int_0^\infty \Phi'(t)\Pr(X \geq t)dt &= \Phi(0) + \int_0^\infty \int_t^\infty \Phi'(t)p(x)dxdt \\
&= \Phi(0) + \int_{\{x \geq t, t \geq 0\}} \Phi'(t)p(x)d\begin{pmatrix} x \\ t \end{pmatrix} \\
&= \Phi(0) + \int_{\{t \leq x, t \geq 0\}} \Phi'(t)p(x)d\begin{pmatrix} x \\ t \end{pmatrix} \\
&= \Phi(0) + \int_0^\infty \int_0^x \Phi'(t)p(x)dtdx \\
&= \Phi(0) + \int_0^\infty \left(\int_0^x \Phi'(t)dt\right)p(x)dx \\
&= \Phi(0) + \int_0^\infty \left(\Phi(x) - \Phi(0)\right)p(x)dx \quad \text{(2nd FTC)} \\
&= \Phi(0) + \int_0^\infty \Phi(x)p(x)dx - \int_0^\infty \Phi(0)p(x)dx \\
&= \Phi(0) + \int_0^\infty \Phi(x)p(x)dx - \Phi(0)\int_0^\infty p(x)dx \\
&= \Phi(0) + \mathbb{E}\left[\Phi(X)\right] - \Phi(0) \\
&= \mathbb{E}\left[\Phi(X)\right]
\end{aligned}
$$

$\square$

**Lemma 2.** *Under the choice of $\Phi_\tau(\cdot)$ above and its associated $\Phi'_\tau(\cdot)$,*

$$\mathbb{E}_{z_1,\ldots,z_m \sim \mathcal{N}(0,I)}\left[\Phi_{\tau_i}(d(\mathbf{x}_i, T_\theta(\mathbf{z}_j)))\right] = \tau_i - \int_{\delta\tau_i}^{\tau_i} \Pr(d(\mathbf{x}_i, T_\theta(\mathbf{z})) < t)dt.$$

*Proof.* By definition, $\Phi_{\tau_i}(0) = \delta\tau_i$.

$$
\begin{aligned}
\mathbb{E}_{z_1,\ldots,z_m \sim \mathcal{N}(0,I)}\left[\Phi_{\tau_i}(d(\mathbf{x}_i, T_\theta(\mathbf{z}_j)))\right] &= \Phi_{\tau_i}(0) + \int_0^\infty \Phi'_{\tau_i}(t)\Pr(d(\mathbf{x}_i, T_\theta(\mathbf{z})) \geq t)dt \quad \text{(Lemma 1)} \\
&= \delta\tau_i + \int_{\delta\tau_i}^{\tau_i} \Pr(d(\mathbf{x}_i, T_\theta(\mathbf{z})) \geq t)dt \\
&= \delta\tau_i + \int_{\delta\tau_i}^{\tau_i} \left(1 - \Pr(d(\mathbf{x}_i, T_\theta(\mathbf{z})) < t)\right)dt \\
&= \delta\tau_i + (\tau_i - \delta\tau_i) - \int_{\delta\tau_i}^{\tau_i} \Pr(d(\mathbf{x}_i, T_\theta(\mathbf{z})) < t)dt \\
&= \tau_i - \int_{\delta\tau_i}^{\tau_i} \Pr(d(\mathbf{x}_i, T_\theta(\mathbf{z})) < t)dt
\end{aligned}
$$

$\square$

**Lemma 3.** *Under the choice of $\Phi_\tau(\cdot)$ above and its associated $\Phi'_\tau(\cdot)$,*

$$\mathcal{L}_{\{\tau_i\}_i}(\theta) = \frac{1}{n}\sum_{i=1}^{n}\frac{1}{mw_i}\sum_{j=1}^{m}\int_{\delta\tau_i}^{\tau_i}\Pr(d(\mathbf{x}_i, T_\theta(\mathbf{z}_j)) < t)dt.$$

*Proof.*

$$\mathcal{L}_{\{\tau_i\}_i}(\theta) = \mathbb{E}_{z_1,\ldots,z_m\sim\mathcal{N}(0,I)}\left[\frac{1}{n}\sum_{i=1}^{n}\frac{1}{w_i}\left(\tau_i - \frac{1}{m}\sum_{j=1}^{m}\Phi_{\tau_i}(d(\mathbf{x}_i, T_\theta(\mathbf{z}_j)))\right)\right]$$

$$= \frac{1}{n}\sum_{i=1}^{n}\frac{1}{w_i}\left(\tau_i - \frac{1}{m}\sum_{j=1}^{m}\mathbb{E}_{z_1,\ldots,z_m\sim\mathcal{N}(0,I)}\left[\Phi_{\tau_i}(d(\mathbf{x}_i, T_\theta(\mathbf{z}_j)))\right]\right)$$

$$= \frac{1}{n}\sum_{i=1}^{n}\frac{1}{w_i}\left(\tau_i - \frac{1}{m}\sum_{j=1}^{m}\left(\tau_i - \int_{\delta\tau_i}^{\tau_i}\Pr(d(\mathbf{x}_i, T_\theta(\mathbf{z}_j)) < t)dt\right)\right) \quad \text{(Lemma 2)}$$

$$= \frac{1}{n}\sum_{i=1}^{n}\frac{1}{mw_i}\sum_{j=1}^{m}\int_{\delta\tau_i}^{\tau_i}\Pr(d(\mathbf{x}_i, T_\theta(\mathbf{z}_j)) < t)dt$$

$\square$

**Equation 3.**

*Proof.*

$$\mathcal{L}_{\{\tau_i\}_i}(\theta) = \mathbb{E}_{z_1,\ldots,z_m\sim\mathcal{N}(0,I)}\left[\frac{1}{n}\sum_{i=1}^{n}\frac{1}{w_i}\left(\tau_i - \frac{m-1}{m}\tau_i - \frac{1}{m}\max(\min_{j\in[m]}d(\mathbf{x}_i, T_\theta(\mathbf{z}_j)), \delta\tau_i)\right)\right]$$

$$= \mathbb{E}_{z_1,\ldots,z_m\sim\mathcal{N}(0,I)}\left[\frac{1}{n}\sum_{i=1}^{n}\frac{1}{w_i}\left(\frac{1}{m}\tau_i - \frac{1}{m}\max(\min_{j\in[m]}d(\mathbf{x}_i, T_\theta(\mathbf{z}_j)), \delta\tau_i)\right)\right]$$

$$= \mathbb{E}_{z_1,\ldots,z_m\sim\mathcal{N}(0,I)}\left[\frac{1}{nm}\sum_{i=1}^{n}\frac{1}{w_i}\left(\tau_i - \max(\min_{j\in[m]}d(\mathbf{x}_i, T_\theta(\mathbf{z}_j)), \delta\tau_i)\right)\right]$$

$\square$

**Equation 4.**

*Proof.*

$$\arg\max_\theta \mathcal{L}_{\{\tau_i\}_i}(\theta) = \arg\max_\theta \mathbb{E}_{z_1,\ldots,z_m\sim\mathcal{N}(0,I)}\left[\frac{1}{nm}\sum_{i=1}^{n}\frac{1}{w_i}\left(\tau_i - \max(\min_{j\in[m]}d(\mathbf{x}_i, T_\theta(\mathbf{z}_j)), \delta\tau_i)\right)\right]$$

$$= \arg\max_\theta \mathbb{E}_{z_1,\ldots,z_m\sim\mathcal{N}(0,I)}\left[\sum_{i=1}^{n}\frac{1}{w_i}\left(\tau_i - \max(\min_{j\in[m]}d(\mathbf{x}_i, T_\theta(\mathbf{z}_j)), \delta\tau_i)\right)\right]$$

$$= \arg\max_\theta \mathbb{E}_{z_1,\ldots,z_m\sim\mathcal{N}(0,I)}\left[\sum_{i=1}^{n}\frac{\tau_i}{w_i} - \sum_{i=1}^{n}\frac{1}{w_i}\max(\min_{j\in[m]}d(\mathbf{x}_i, T_\theta(\mathbf{z}_j)), \delta\tau_i)\right]$$

$$= \arg\max_\theta \mathbb{E}_{z_1,\ldots,z_m\sim\mathcal{N}(0,I)}\left[-\sum_{i=1}^{n}\frac{1}{w_i}\max(\min_{j\in[m]}d(\mathbf{x}_i, T_\theta(\mathbf{z}_j)), \delta\tau_i)\right]$$

$$= \arg\min_\theta \mathbb{E}_{z_1,\ldots,z_m\sim\mathcal{N}(0,I)}\left[\sum_{i=1}^{n}\frac{1}{w_i}\max(\min_{j\in[m]}d(\mathbf{x}_i, T_\theta(\mathbf{z}_j)), \delta\tau_i)\right]$$

$\square$

**Lemma 4.** *Suppose $p_\theta$ is continuous at all data points $\mathbf{x}_1, \ldots, \mathbf{x}_n$, under the choice of $w_i = \int_{\delta\tau_i}^{\tau_i} \mathrm{vol}(B_t(\mathbf{x}_i))dt := \int_{\delta\tau_i}^{\tau_i} \int_{B_t(\mathbf{x}_i)} d\mathbf{x}dt$, where $B_r(\mathbf{x}) = \{\mathbf{y}|d(\mathbf{y}, \mathbf{x}) < r\}$ is an open ball of radius $r$ centred at $\mathbf{x}$,*

$$\lim_{\{\tau_i \to 0^+\}_i} \mathcal{L}_{\{\tau_i\}_i}(\theta) = \frac{1}{n} \sum_{i=1}^n p_\theta(\mathbf{x}_i)$$

*Proof.*

$$\mathcal{L}_{\{\tau_i\}_i}(\theta) = \frac{1}{n} \sum_{i=1}^n \frac{1}{mw_i} \sum_{j=1}^m \int_{\delta\tau_i}^{\tau_i} \Pr(d(\mathbf{x}_i, T_\theta(\mathbf{z}_j)) < t)dt \quad \text{(Lemma 3)}$$

$$= \frac{1}{n} \sum_{i=1}^n \frac{1}{mw_i} \sum_{j=1}^m \int_{\delta\tau_i}^{\tau_i} \int_{B_t(\mathbf{x}_i)} p_\theta(\mathbf{x})d\mathbf{x}dt$$

$$= \frac{1}{nm} \sum_{i=1}^n \sum_{j=1}^m \frac{1}{w_i} \int_{\delta\tau_i}^{\tau_i} \int_{B_t(\mathbf{x}_i)} p_\theta(\mathbf{x})d\mathbf{x}dt$$

$$= \frac{1}{nm} \sum_{i=1}^n \sum_{j=1}^m \frac{\int_{\delta\tau_i}^{\tau_i} \int_{B_t(\mathbf{x}_i)} p_\theta(\mathbf{x})d\mathbf{x}dt}{\int_{\delta\tau_i}^{\tau_i} \int_{B_t(\mathbf{x}_i)} d\mathbf{x}dt}$$

$$\lim_{\{\tau_i \to 0^+\}_i} \mathcal{L}_{\{\tau_i\}_i}(\theta) = \frac{1}{nm} \sum_{i=1}^n \left( \lim_{\tau_i \to 0^+} \left( \sum_{j=1}^m \frac{\int_{\delta\tau_i}^{\tau_i} \int_{B_t(\mathbf{x}_i)} p_\theta(\mathbf{x})d\mathbf{x}dt}{\int_{\delta\tau_i}^{\tau_i} \int_{B_t(\mathbf{x}_i)} d\mathbf{x}dt} \right) \right)$$

$$= \frac{1}{nm} \sum_{i=1}^n \sum_{j=1}^m \left( \lim_{\tau_i \to 0^+} \frac{\int_{\delta\tau_i}^{\tau_i} \int_{B_t(\mathbf{x}_i)} p_\theta(\mathbf{x})d\mathbf{x}dt}{\int_{\delta\tau_i}^{\tau_i} \int_{B_t(\mathbf{x}_i)} d\mathbf{x}dt} \right)$$

$$= \frac{1}{nm} \sum_{i=1}^n \sum_{j=1}^m \left( \lim_{\tau_i \to 0^+} \frac{\int_{B_{\tau_i}(\mathbf{x}_i)} p_\theta(\mathbf{x})d\mathbf{x} - \delta \int_{B_{\delta\tau_i}(\mathbf{x}_i)} p_\theta(\mathbf{x})d\mathbf{x}}{\int_{B_{\tau_i}(\mathbf{x}_i)} d\mathbf{x} - \delta \int_{B_{\delta\tau_i}(\mathbf{x}_i)} d\mathbf{x}} \right) \quad \text{(L'Hôpital and 2nd FTC)}$$

$$= \frac{1}{nm} \sum_{i=1}^n \sum_{j=1}^m \left( \lim_{\tau_i \to 0^+} \frac{\int_{B_{\tau_i}(\mathbf{x}_i)} p_\theta(\mathbf{x})(1 - \delta \mathbf{1}_{B_{\delta\tau_i}(\mathbf{x}_i)}(\mathbf{x}))d\mathbf{x}}{\int_{B_{\tau_i}(\mathbf{x}_i)} 1 - \delta \mathbf{1}_{B_{\delta\tau_i}(\mathbf{x}_i)}(\mathbf{x})d\mathbf{x}} \right)$$

$$= \frac{1}{nm} \sum_{i=1}^n \sum_{j=1}^m \left( \lim_{\tau_i \to 0^+} \frac{\int_0^{\tau_i} (1 - \delta \mathbf{1}_{\{r < \delta\tau_i\}}(r)) \int_{\{\mathbf{x}|d(\mathbf{x},\mathbf{x}_i)=r\}} p_\theta(\mathbf{x})d\mathbf{x}dr}{\int_0^{\tau_i} (1 - \delta \mathbf{1}_{\{r < \delta\tau_i\}}(r)) \int_{\{\mathbf{x}|d(\mathbf{x},\mathbf{x}_i)=r\}} d\mathbf{x}dr} \right)$$

$$= \frac{1}{nm} \sum_{i=1}^n \sum_{j=1}^m \left( \lim_{\tau_i \to 0^+} \frac{\int_{\{\mathbf{x}|d(\mathbf{x},\mathbf{x}_i)=\tau_i\}} p_\theta(\mathbf{x})d\mathbf{x}}{\int_{\{\mathbf{x}|d(\mathbf{x},\mathbf{x}_i)=\tau_i\}} d\mathbf{x}} \right) \quad \text{(L'Hôpital and 2nd FTC)}$$

$$= \frac{1}{nm} \sum_{i=1}^n \sum_{j=1}^m p_\theta(\mathbf{x}_i) \quad \text{(Continuity of } p_\theta)$$

$$= \frac{1}{n} \sum_{i=1}^n p_\theta(\mathbf{x}_i)$$

$\square$

Note that under common metrics like $\ell_p$ distances, $w_i$ can be found in closed form, i.e., $\mathrm{vol}(B_t(\mathbf{x}_i)) = (2t)^d \frac{\Gamma(1+1/p)^d}{\Gamma(1+d/p)}$, and so $w_i = \int_{\delta\tau_i}^{\tau_i} \mathrm{vol}(B_t(\mathbf{x}_i))dt = \int_{\delta\tau_i}^{\tau_i} (2t)^d \frac{\Gamma(1+1/p)^d}{\Gamma(1+d/p)}dt = \frac{(2(1-\delta)\tau_i)^{d+1}}{2(d+1)} \cdot \frac{\Gamma(1+1/p)^d}{\Gamma(1+d/p)}$, where $\Gamma(\cdot)$ denotes the gamma function.

## B  PSEUDO CODE FOR IMLE

---

**Algorithm 2** Implicit maximum likelihood estimation (IMLE) procedure

---

**Require:** The set of inputs $\{\mathbf{x}_i\}_{i=1}^n$
  1: Initialize the parameters $\theta$ of the generator $T_\theta$
  2: **for** $k = 1$ **to** $K$ **do**
  3:      Pick a random batch $S \subseteq [n]$
  4:      Draw latent codes $Z \leftarrow \mathbf{z}_1, ..., \mathbf{z}_m$ from $\mathcal{N}(0, \mathbf{I})$
  5:      $\sigma(i) \leftarrow \arg\min_{j \in [m} d(\mathbf{x}_i, T_\theta(\mathbf{z_j})) \; \forall i \in S$
  6:      **for** $l = 1$ **to** $L$ **do**
  7:          Pick a random mini batch $\tilde{S} \subseteq S$
  8:          $\theta \leftarrow \theta - \eta \nabla_\theta \left( \sum_{i \in \tilde{S}} d\left(\mathbf{x}_i, T_\theta\left(\mathbf{z}_{\sigma(\mathbf{i})}\right)\right) \right) / |\widetilde{S}|$
  9:      **end for**
 10: **end for**
 11: **return** $\theta$

---

