# OpenReview forum: "Adaptive IMLE for Few-shot Image Synthesis"
_ICLR.cc/2023/Conference — Submitted to ICLR 2023_

### Official Review · Reviewer_uhYC · 2022-10-22

**Confidence:** 4
**Correctness:** 3
**Technical Novelty And Significance:** 4
**Empirical Novelty And Significance:** Not applicable
**Recommendation:** 6

**Clarity, Quality, Novelty And Reproducibility:**

The clarity on writing should be further improved. In my opinion, the design of the adaptive IMLE is novel.


**Strength And Weaknesses:**

Strength:
1. The idea of introducing an adaptive radii for IMLE is interesting, which increase the degree of the vanilla IMLE.
2. Leveraging lemma 1 to construct the adaptive is very clever. It provides a very useful tool to depict the distribution of the distance between the training example and a generated sample.
3. The visual results tend to prove that this adaptive objective could produce more diverse example.

Weaknesses:
1. The written of this paper is not very clear. Beginning from lemma 3, the notation $w_i$ is frequently appeared. But there isn't any explanation about this notation, and why does introduce $w_i$?
2. I don't understand the definition of $L_{\{\tau_i\}_i}$ in the first line of the proof for lemma 3 in appendix. If possible, please provide more details.
3. The number of sample $m$ should play an important role in this work. Even this work design the curriculum strategy to avoid increasing $m$. It would be better to offer some results under different values of $m$.
4. Comparing with the vanilla IMLE, why does the proposed objective achieve better results? This paper does not clearly claim this point.

**Summary Of The Paper:**

This work designs a more general IMLE objective by introducing an adaptive neighbourhood radii. The theoretical derivation and the proposed curriculum learning strategy are beautiful and impressive. Some experiments are also conducted to verify the effectiveness of the proposed method.

**Summary Of The Review:**

This work devises a theoretically sound objective to generalize the IMLE, and achieves stable and better results. Even though the writing of this paper should be improved, it is still a beautiful work.

---

> ### Author Response · Authors · 2022-11-20
> **Response to Review uhYC**
>
> ### Q1: Missing explanation about $w_i$ beginning from lemma 3, why introduce $w_i$?
> A1: $w_i$ was first introduced in Eqn 2, but as mentioned in the paragraph beneath it, we choose the concrete value for it later based on the insight revealed by later lemmas. That happened in Lemma 4, when the reason for having $w_i$ becomes clear. Namely, $w_i$ is a normalizing factor that allows each term to be interpreted as the average probability per unit area, which converges to the probability density as the size of the neighbourhood goes to zero. $w_i$ is the volume of the neighbourhood.
>
> ### Q2: I don’t understand $L_\tau$ in the first line of the proof for lemma 3 in the appendix, please explain.
> A2: $L_\tau$ was defined in Eqn. 2, which is the objective function we propose in our generalized formulation. Intuitively, the objective tries to measure the weighted sum of the densities around all the data points.
>
> ### Q3: It would be better to show some results under different values of m
> A3: As suggested, we have compared training progress for different values of $m$ and added the results to the supplementary material*. As shown, a larger $m$ could speed up convergence in terms of the number of iterations at the cost of slower per-iteration speed. In practice, we tune $m$ to strike a good balance between convergence rate and iteration speed.
>
> \* Due to technical difficulties we encountered when uploading the supplementary material to OpenReview, the version of supplementary material that shows up is still the old version. We reached out to the PCs and are waiting for them to help us update it. Meanwhile, please refer to the following anonymous GitHub page to access the supplementary material: https://github.com/iclr-2023-paper-5926/iclr2023-paper-5926
>
> ### Q4: Why does the proposed objective achieve better results than vanilla IMLE?
> A4: The proposed objective can better adapt to data points with varying difficulties than vanilla IMLE. It cansimultaneously avoid overfitting to easier data points and underfitting to harder data points. We added a more concrete explanation to the last paragraph of Section 3.

---

> > ### Comment · Reviewer_uhYC · 2022-11-23
> > **Feedback to authors**
> >
> > The response has addressed most of my issues. In particular, Fig. 5, 6, and 7 verify the advantages of the proposed method. I suggest these figures should be added into the paper, at least in appendix.
> >
> > I thus will remain my positive score.

---

> > > ### Author Response · Authors · 2022-11-23
> > > **Thank you for the response**
> > >
> > > Thank you for the response and the suggestions. We will add those figures and accompanying discussion to the camera-ready.

---

### Official Review · Reviewer_xM5B · 2022-10-25

**Confidence:** 3
**Correctness:** 3
**Technical Novelty And Significance:** 3
**Empirical Novelty And Significance:** 3
**Recommendation:** 6

**Clarity, Quality, Novelty And Reproducibility:**

This work's presentation of proposed framework is clear and easy to follow. It is a more generalized version of a previous method. Reproducibility is upon the code release though the algorithm of each step is already given.

**Details Of Ethics Concerns:**

Not aware of concerns.

**Strength And Weaknesses:**

+ Strict prove on theoretical guarantees of the generalized formulation hold under weaker condition
+ A systematic level up of vanilla IMLE algorithm
+ Obvious improvement of results under few-shot setting

A good work about making GAN training less data-hungry and potentially using less compute resources, which is applicable to personalized generation with a small library of data. Please find my minor concerns below:

To understand how the number of training data is affecting the performance, it is better to conduct an experiment by ablating this hyper-parameter and show a curve about how FID will increase with less training examples.

Just curious what is the smallest number of images that the proposed method could work? 100? 10? 1? The previous transferring learning based methods, though requiring pretraining on an external data, are more aligned with human learning process of learning new knowledge quickly after learning old knowledge and show even 1-shot setting is possible. The proposed method belongs to learning from scratch with a few amount of data. Is it limited in for example just learning 100 faces of one person (obama) instead of learning 100 faces of different persons? Essentially, GAN should be a data engine to benefit more downstream tasks. If the generated is still less diverse, it may not be quite useful as compared to transfer learning approaches.

It is suggested to include some results on non-facial data as face has been heavily explored, in order to see how robust the algorithm is cross different type of datasets.


**Summary Of The Paper:**

This paper enables training GANs with a few amount of data from scratch. It is built on top of an existing method IMLE which is improved to adaptive IMLE to avoid model collapse or model overfitting. The more generalized form of IMLE is presented which covers the original formulation as a special case. Extensive experiments on different datasets are conducted to show that the proposed method could generate higher quality and more diverse results.

**Summary Of The Review:**

A new perspective to tackle the few shot generation problem. Good work but I worried a bit about its performance over other non-facial dataset and eventually how useful it will be compared to transfer learning setting.

---

> ### Author Response · Authors · 2022-11-20
> **Response to Reviewer xM5B**
>
> ### Q1: What is the effect on the performance when lowering the number of training data?
> A1: As suggested, we trained our model on the Oxford Flowers dataset with only 10 random images. Our method converges very quickly and generates high-quality images which can be found on the last page of the supplementary material.
>
> \* Due to technical difficulties we encountered when uploading the supplementary material to OpenReview, the version of supplementary material that shows up is still the old version. We reached out to the PCs and are waiting for them to help us update it. Meanwhile, please refer to the following anonymous GitHub page to access the supplementary material: https://github.com/iclr-2023-paper-5926/iclr2023-paper-5926
>
> ### Q2: Is the proposed method limited to learning 100 faces of one person instead of learning 100 faces of different people?
> A2: In fact, the FFHQ subset consists of faces from different people, and the results show that our method does well on it.
>
> ### Q3: It is suggested to include some results on non-facial data as faces have been heavily explored.
> A3: As suggested, we applied our method to the Oxford Flowers dataset on the 10-shot and 100-shot settings. The results are included on the last page of the supplementary material. This demonstrates that our method works on non-facial datasets as well.

---

> > ### Comment · Reviewer_xM5B · 2022-11-23
> > **Feedback to authors**
> >
> > Thanks for the feedback that addresses the concerns. To me the transfer learning still better aligns the logic of how human learn new knowledge quickly. Results with 10 data or even fewer looks far from that obtained by transfer learning methods. But I'm not opposed to learning with a few data from scratch which has still some value. Authors are suggested to verify the proposed idea over more types of data in the future (not just face and flower), considering the generation field is moving fast towards more and more higher resolution and complexity of data. I'll keep my original score.

---

> > > ### Author Response · Authors · 2022-11-25
> > > **Thank you for the response**
> > >
> > > Thank you for the response and the suggestions. There are some situations where transfer learning would be helpful, but there are also other situations (e.g., when no auxiliary dataset is available) that necessitate training from scratch. There is certainly the potential for applying our method to model other kinds of data, which we will explore in future work as suggested.

---

### Official Review · Reviewer_owTn · 2022-10-29

**Confidence:** 4
**Correctness:** 3
**Technical Novelty And Significance:** 2
**Empirical Novelty And Significance:** 2
**Recommendation:** 6

**Clarity, Quality, Novelty And Reproducibility:**

**Clarity.** The clarity of the paper can be improved. For instance, a more comprehensive (theoretical) explanation of the limitation of IMLE and a comparison between IMLE and the proposed method would be helpful.

**Quality.** While the paper performs the experiments on diverse datasets, yet needs to include recent baselines that outperform the presented method.

**Novelty.** The proposed method seems novel, yet it could be clearer due to not very good clarity.

**Reproducibility.** The supplementary material contains the code for reproducing the results, indicating good reproducibility.

**Strength And Weaknesses:**

### Strengths
* The problem setup of few-shot image synthesis is practical and interesting.
* The paper performs extensive experiments across diverse datasets.
* The paper shows solid experimental results, both quantitatively and qualitatively.

### Weaknesses
* While the problem definition (few-shot image synthesis) is interesting in general, I am not convinced about the detailed problem setup in not using auxiliary datasets. An explanation highlighting the importance of such a setup or comparing the results with fine-tuning-based methods would be welcome.
* The paper is hard to follow. Considering that the paper proposes the generalized version of IMLE, the clarity of the paper would be much better if it explicitly mentioned the limitations of IMLE (at least in brief) in the theoretical aspect. However, the paper only mentions that “restrictive conditions need to be satisfied for the theoretical guarantees of IMLE to hold, such as requiring a uniform optimal likelihood for all data points”. Why, what, and how are such conditions restrictive, while the proposed method is not?
*  I think the ablation study (Table 3) should be plotted (instead of Table) to compare IMLE and adaptive IMLE. Few-shot synthesis problems are generally vulnerable to the overfitting of the generator. As the proposed adaptive IMLE objective includes additional regularizations (e.g., $\min$ with $\delta \tau_i >0$)than IMLE, hence more difficult to optimize, I wonder whether the performance gain is simply from alleviating the overfitting or from indeed “better” objective design. At least, I strongly recommend authors put more details on how the ablation study is performed (e.g., “best” performance during the training is reported); but still, showing plots with respect to number of iterations would be much more welcome.
* While the proposed method shows superior performance among the reported baselines, the current manuscript misses some recent baselines (even outperforming adaptive IMLE), e.g., [1]. At least, a discussion with this baseline would strengthen the manuscript more.

[1] FakeCLR: Exploring Contrastive Learning for Solving Latent Discontinuity in Data-Efficient GANs, ECCV 2022. (arXiv: Jul 2022)

### Minor comment
* The equation number is not stated in the sentence between Eqn. 3 and Eqn. 4.

### Questions
* Why does the precision of IMLE be better than the proposed Adaptive IMLE?
* The proposed method can be applied to any few-shot synthesis problem; do the authors try to apply the method in different domains?

**Summary Of The Paper:**

The paper proposes a new objective for the problem of few-shot image synthesis based on extending implicit maximum likelihood estimation (IMLE). Specifically, they derive a data-adaptive IMLE objective for training the implicit generative model. Experiment results on six few-shot image synthesis datasets, namely Grumpy Cat, Obama, Cat, Panda, Dog, and the subset of FFHQ, demonstrate the proposed method outperforms the previous few-shot image synthesis methods.

**Summary Of The Review:**

This paper tackles the few-shot image synthesis problem by extending the IMLE objective to be “image-adaptive”. However, I feel the overall clarity could be improved. For the evaluation side, I am not fully convinced due to the missing baseline and the concerns about ablation studies. In these respects, I am currently on the negative side.

---

> ### Author Response · Authors · 2022-11-20
> **Response to Reviewer owTn**
>
> ### Q1: Why consider this problem setting (few-shot image synthesis without auxiliary datasets)?
> A1: This problem setting was proposed and studied in prior work [a,b], and is not new. It is most important when there is no large-scale auxiliary dataset that is semantically similar to the task at hand. Leveraging a semantically dissimilar auxiliary dataset can introduce bias due to the large domain gap between it and the task at hand.
>
> ### Q2: The paper will be clearer if it explicitly mentions the theoretical limitations of IMLE
> A2: We have updated Section 3 (last paragraph) in the manuscript with more details on the theoretical limitations and why they are restrictive.
>
> ### Q3: Ablation study should include more details on how IMLE and adaptive IMLE is compared, preferably showing plots w.r.t. iteration.
> A3: As suggested, we added plots showing FID and loss comparisons between Adaptive IMLE and vanilla IMLE at different iterations to Sect C of the supplementary material. As shown, Adaptive IMLE performs significantly better across different iterations.
>
> \* Due to technical difficulties we encountered when uploading the supplementary material to OpenReview, the version of supplementary material that shows up is still the old version. We reached out to the PCs and are waiting for them to help us update it. Meanwhile, please refer to the following anonymous [GitHub page](https://github.com/iclr-2023-paper-5926/iclr2023-paper-5926) to access the supplementary material: https://github.com/iclr-2023-paper-5926/iclr2023-paper-5926
>
> ### Q4: Missing baseline FakeCLR [c]
> A4: Thanks for bringing this to our attention. The mentioned paper is published at ECCV 2022 and is therefore considered concurrent work as per ICLR policy (last Q&A in https://iclr.cc/Conferences/2023/ReviewerGuide), so technically the comparison to this baseline is not required. Below is the relevant excerpt of ICLR policy:
> “We consider papers contemporaneous if they are published (available in online proceedings) within the last four months. That means, since our full paper deadline is September 28, if a paper was published (i.e., at a peer-reviewed venue) on or after May 28, 2022, authors are not required to compare their own work to that paper.”
> Nevertheless, we tried training the baseline on our datasets, but it failed to converge with the given hyperparameter settings in their repository. We also reached out to the authors and got suggestions on hyperparameter tuning, but the model performance still did not improve after several days of training. We will, however, cite and discuss the paper in the camera-ready.
>
> ### Q5: Why is the precision of IMLE better than the proposed Adaptive IMLE?
> A5: Note that precision of Adaptive IMLE is only very slightly (0.1%-0.6%) worse than vanilla IMLE, which is insignificant compared to the improvement it achieves on recall (39.3%-49.0%). In the few-shot setting, the precision metric from [d] is not very meaningful because it regards a generated sample as “precise” if it falls within the k-nearest neighbourhood of a real image. Because there are few real images in the few-shot setting, different real images are relatively dissimilar from one another. So, the k-nearest neighbourhood is relatively large, and so both relatively low and high-quality samples can be considered “precise”. Vanilla IMLE generates similar lower quality samples, whereas Adaptive IMLE generates diverse higher quality samples. Because the samples from the latter are more diverse, a few samples can occasionally be considered not to be “precise” and lead to a very slight decrease in precision.
>
> ### References
> [a] Bingchen Liu, Yizhe Zhu, Kunpeng Song, and Ahmed Elgammal. Towards faster and stabilized GAN training for high-fidelity few-shot image synthesis.
>
> [b] Chaerin Kong, Jeesoo Kim, Donghoon Han, and Nojun Kwak. Smoothing the generative latent space with mixup-based distance learning. In European Conference on Computer Vision, 2022.
>
> [c] FakeCLR: Exploring Contrastive Learning for Solving Latent Discontinuity in Data-Efficient GANs, ECCV 2022. (arXiv: Jul 2022)
>
> [d] Tuomas Kynkäänniemi, Tero Karras, Samuli Laine, Jaakko Lehtinen, and Timo Aila. Improved precision and recall metric for assessing generative models. Advances in Neural Information Processing Systems, 32, 2019.

---

> > ### Comment · Reviewer_owTn · 2022-11-22
> > **Response**
> >
> > Thank you for your thoughtful response. I think the response addressed most of my concerns. Here are a few remarks:
> >
> > - I understand the problem setup was proposed and studied in prior work [a,b], and is not new, but still, it would be much better if it is explicitly explained in the manuscript (e.g., Introduction).
> > - I also understand my suggested baseline can be regarded as concurrent work. However, irrespective of the acceptance of the paper, I strongly believe the discussion with them should be included in the camera-ready version for better future research. Please add the discussion if the paper is accepted.
> >
> > I will raise my score to 6.

---

> > > ### Author Response · Authors · 2022-11-22
> > > **Thank you for the response**
> > >
> > > Thank you for the response. We will add the discussion on the problem setup and the mentioned baseline in the camera-ready.

---

### Official Review · Reviewer_jG1N · 2022-10-30

**Confidence:** 4
**Correctness:** 4
**Technical Novelty And Significance:** 2
**Empirical Novelty And Significance:** Not applicable
**Recommendation:** 3

**Clarity, Quality, Novelty And Reproducibility:**

This paper provides a theoretical analysis of IMLE by providing a generalized formulation, which is important.
The pseudo-code is provided for reproducibility.

**Strength And Weaknesses:**

Strength:
1.	Formulating a generalized form of techniques is essential for further development, and the few-shot learning setting is specifically studied.
2.	Clear paper writing about background introduction and method derivation & development.
3.	Consistent performance gain and promising results. Visualization are also provided.

Weakness:
1.	My biggest concern is regarding ignoring w_i in the Eq.5. The explanation seems incomplete (the unweighted objective, since tau_i and there fore w_i is fixed). Please double-check and clearly explain why w_i can be ignored.
2.	Why Prec drops slightly while FID and Rec. are improved largely? It is necessary to explain the reason. Acturally, Prec are very high and I am not sure whether it Is a good metric.
3.	Training details should be provided. Do you need additional large-scale data for pre-training? Also, do you need to find the nearest generated samples for all training data within each batch? If yes, the memory usage is pretty large and I wonder whether your approach can address this issue.
4.	The comparison between IMLE and adaptative IMLE is only on Obama and FFHQ while more experimental comparisons are performed in Table 1-2. Can you explain the reason? From my perspective, the comparison with vanilla IMLE is much more critical.
5.	For the optimization of IMLE, one weakness is that the neighbor search is centered w.r.t. each training data, then, it is possible that a very limited number of generated samples are selected in one batch. Will that be risky, and do your methods can mitigate this issue also?


**Summary Of The Paper:**

This paper proposed a generalized formulation of IMLE, Adaptive IMLE, for few-shot image synthesis. As the generator trained in GAN suffers from the mode collapse issue, i.e., GAN may overfit to a subset of training samples. Then, IMLE is proposed to avoid this issue by optimizing training images to have some similar generated images. However, as strict restrictions are applied on IMLE and may impede training efficiency in the few-shot setting, this paper proposed Adaptive IMLE by adjusting the radius w.r.t. each training data. Detailed theoretical analysis is provided, and the experimental study demonstrates its effectiveness. However, the discussion is very limited.

**Summary Of The Review:**

The discussion is very limited. Please see my comments above.

---

> ### Author Response · Authors · 2022-11-20
> **Response to Reviewer jG1N**
>
> ### Q1: Why $w_i$ can be ignored
> A1: We have added a proof to Sect D of the supplementary material* that shows optimizing the unweighted objective is equivalent to optimizing the weighted one.
>
> \* Due to technical difficulties we encountered when uploading the supplementary material to OpenReview, the version of supplementary material that shows up is still the old version. We reached out to the PCs and are waiting for them to help us update it. Meanwhile, please refer to the following anonymous [GitHub page](https://github.com/iclr-2023-paper-5926/iclr2023-paper-5926) to access the supplementary material: https://github.com/iclr-2023-paper-5926/iclr2023-paper-5926
>
> ### Q2: Why is the precision so high?
> A2: The precision metric from [c] calculates “the probability that a randomly generated image $\phi_g$ falls within the support of the distribution of the real images $P_r$”. A generated image $\phi_g$ is considered “within the support of $P_r$” if there exists a real image $\phi_r \in \Phi_r$, such that $||\phi_g - \phi_r||_2 \leq || \phi_r - NN_k(\phi_r, \Phi_r)||_2$, where $\Phi_r$ is the set of all real images and $NN_k(\phi_r, \Phi_r)$ is the k-nearest neighbour of $\phi_r$ in $\Phi_r$. In the few-shot setting, the real images tend to be more dissimilar, meaning that the RHS $|| \phi_r - NN_k(\phi_r, \Phi_r)||_2$ would tend to be big.
> ### Q3: Why the precision dropped slightly when FID and recall improved largely? I am not sure whether precision is a good metric.
> A3: We agree that precision is not a good metric in the few-shot setting; we reported it for completeness following prior works [a,b]. As discussed in A2, the precision metric from [c] regards a generated sample as “precise” if it falls within the k-nearest neighbourhood of a real image. Because there are few real images in the few-shot setting,  different real images are relatively dissimilar from one another. So, the k-nearest neighbourhood is relatively large, and so the quality of generated images must be quite bad for it to be reflected in the precision metric. Therefore, precision is not very effective for measuring image quality.
>
> On the other hand, FID and recall metrics are more meaningful, and hence better reflect the performance improvements achieved by our method. FID does not rely on the k-nearest neighbourhood for evaluating sample quality. Instead, FID directly compares the distributions of the generated images and the real images. Therefore, FID is more effective in measuring image quality compared to the precision metric from [c]. The recall metric from [c] is in some sense the opposite of precision, and measures mode coverage. It considers a real image to be covered if it falls within the k-nearest neighbourhood of some generated image. Since we can generate arbitrarily many images, the k-nearest neighbourhood of a generated image is small, and so the recall metric from [c] is still meaningful and sensitive to mode coverage.
>
> ### Q4: Do you need additional large-scale data for pre-training?
> A4: No, our method is trained on a small dataset from scratch in all experiments. We do not pre-train our model on any other dataset.
>
> ### Q5: Do you need to find the nearest generated samples for all training data within each batch and how would you handle the memory usage?
> A5: The different training examples find the nearest neighbour from a shared pool of samples, and we fix the size of the shared pool to a manageable scale.
> ### Q6: Comparison between IMLE and adaptive IMLE is only on Obama and FFHQ.
> A6: We selected those two datasets because they are among the more challenging ones. On the other four datasets (dog, cat, panda and grumpy cat), Adaptive IMLE also outperforms vanilla IMLE (e.g., it improves FID by 18%, 13%, 2%, and 10% respectively).
>
> ### Q7: In IMLE, would it be risky that only a limited number of generated samples are selected in one batch for optimization? Does adaptive IMLE mitigate this issue?
> A7: At the start of each optimization problem, we find the nearest neighbour of data examples within a pool of generated samples. This pool of randomly generated samples is large enough so that each data example ends up with a reasonable and different nearest neighbour. However, in case two data examples end up with the same nearest neighbour, we perturb the selected latent code values by adding a small noise.
> ### References
> [a] Bingchen Liu, Yizhe Zhu, Kunpeng Song, and Ahmed Elgammal. Towards faster and stabilized GAN training for high-fidelity few-shot image synthesis.
>
> [b] Chaerin Kong, Jeesoo Kim, Donghoon Han, and Nojun Kwak. Smoothing the generative latent space with mixup-based distance learning. In European Conference on Computer Vision, 2022.
>
> [c] Tuomas Kynkäänniemi, Tero Karras, Samuli Laine, Jaakko Lehtinen, and Timo Aila. Improved precision and recall metric for assessing generative models. Advances in Neural Information Processing Systems, 32, 2019.

---

> > ### Comment · Reviewer_jG1N · 2022-11-30
> > **Thank you for your response**
> >
> > Thank you for your detailed explanation. However, I think the response is not quite complete.
> >
> > For Q1: I have read the derivation in your new supp. I do acknowledge that w_i can be found in closed form. However, how does it relate to ignoring w_i in your Eq. 5. Or do you mean the w_i can be pre-calculated and then fixed during your training? From my perspective, the model obtained by separating two optimization objectives may not be the same as optimizing the two objectives jointly. As such, I think there is still some gap between your derivation and your conclusion indicated in Eq. 5. Meanwhile, I do not see any edits in the main paper, at least for what I have raised in my previous comment.
> >
> > For Q5: Please explain them clearly. For your so-called "managable" size, how large it is.
> >
> > For the other questions, I think I am good now. I can raise my score to 4 (though it is not provided) and I would like to encourage the authors to share extra thoughts about Q1.

---

> > > ### Author Response · Authors · 2022-12-01
> > > **Thank you for the response**
> > >
> > > Thank you for the reply and the feedback.
> > >
> > > Q1: In section D of the supplementary material we prove that the set of solutions for the unweighted objective is equal to the set of solutions for the weighted one — this is true regardless of whether $w_i$’s can be found in closed form or not. This means that optimizing the unweighted objective (as in Eqn. 5) would arrive at the same solution as optimizing the weighted objective (as in Eqn.4), and so we could safely ignore $w_i$.
> > >
> > > As for the edits in the main paper, we have made substantial edits to the supplementary material but couldn’t do so for the main paper because of the difficulty of making the content fit within the page limit.
> > >
> > > Q5: The shared pool size is a hyperparameter that can be tuned based on the available resources. In our experiments, the pool contains ~10k samples of 256x256 resolution which only use less than 10GB of memory. Empirically, we found that this shared pool size strikes a good balance between training speed and quality.
> > >
> > > We hope our response addressed your concerns and if you have any remaining concerns, we would be more than happy to clarify them.

---

### Official Review · Reviewer_Me9N · 2022-10-31

**Confidence:** 3
**Correctness:** 3
**Technical Novelty And Significance:** 2
**Empirical Novelty And Significance:** 3
**Recommendation:** 6

**Clarity, Quality, Novelty And Reproducibility:**


**Clarity:**
Overall, I found the paper fairly easy to read, but had some questions regarding some of the proposed theory and derivations:
- There is some handwaving in the derivation from equation (3) to (4). In particular, $\tau_i$ is chosen to depend on $T_{\theta}$ prior to equation (3), but then the $\tau_i$ term is dropped in the loss function in equation (4). Can we truly treat $\tau_i$ as a constant here?
- The derivation in Lemma 2 appears for a fixed sample $z_j$, rather than an expectation over the empirical distribution.
- There are no assumptions placed on the distribution $p_{\theta}$, which makes it hard to verify some of the results. For example, why does the limit in the third to last line in the proof of Lemma 4 exist? At a minimum, one would require continuity of $p_{\theta}$ at the datapoints to assert this.

**Novelty and significance:**
To the reviewer’s knowledge, this extension of IMLE appears novel, but it could be that I am unaware of certain prior work. The experimental results also appear quite solid.

**General comments:**
- Ultimately, the loss that ends up being used by the authors is equation (5) versus the original IMLE equation (1). The only difference here is in taking the maximum between the distance and the threshold $\delta \tau_i$. The weights, $w_i$, however, can be computed in closed-form for certain metrics, such as the distance induced by an $\ell_p$-norm. Empirically, did it have an impact on performance if the loss with weighting $w_i$ was used versus equation (5)?
- Along these lines, it would be nice to see some low-dimensional examples showcasing how the proposed loss actually helps in “adapting” to difficult examples. Figure 1 provides some intuition, but it would be nice to see some 2- or 3-dimensional experiments showing this more concretely.
- Since the framework allows flexibility in choosing a distance metric $d(\cdot,\cdot)$, did the authors try different metrics other than $\ell_2$-distance, such as an LPIPS metric? Did this influence performance at all?
- There is a missing equation reference in the middle of page 5

**Strength And Weaknesses:**


**Strengths:**
- The new adaptive loss function is principled in that it can be shown in a limiting sense to converge to a maximum likelihood-based objective.
- The empirical results show quantitative and qualitative improvements in terms of generation quality and diversity of the images. To the reviewer's knowledge, comparisons against strong few-shot image synthesis baselines are considered for comparison.

**Weaknesses:**
- Some notions of the theory would be impractical to implement, such as choosing the correct weights $w_i$ in the optimization for certain distance metrics, which requires computing the volume of a neighborhood around a given datapoint.
- There are some issues in the clarity/derivations that would be good to improve upon.


**Summary Of The Paper:**

This paper considers the problem of few-shot image synthesis, i.e., learning to generate samples from a distribution given only few examples. A previous approach, dubbed Implicit Maximum Likelihood Estimation (IMLE), showed that one perform few-shot synthesis by minimizing a loss that penalizes the distance between generated samples and true samples. This loss was shown to exhibit nice properties, such as convergence to the true likelihood, but operated under strong assumptions (e.g., all training images have the same likelihood). This work builds upon IMLE by proposing an adaptive version that is able to weigh examples from the training set based on their difficulty and theoretically show that their loss is a generalization of IMLE operating under weaker assumptions. Results on few-shot image synthesis benchmarks show that learning with the Adaptive IMLE can improve generation.

**Summary Of The Review:**

Overall, I think that the paper presents a nice extension of previous work in order to improve few-shot image synthesis. The experimental results seem promising, but there is some lack of polish in the presentation of the material. I am leaning more on the positive-side based on the results.

---

> ### Author Response · Authors · 2022-11-20
> **Response to Reviewer Me9N**
>
> ### Q1: $w_i$ may be impractical to implement for certain distance metric
> A1: $w_i$ does not need to be computed explicitly, since the weighted (with $w_i$) and unweighted (without $w_i$) objectives have the same optimizers. We added a proof Sect D of the supplementary material* to show this.
>
> \* Due to technical difficulties we encountered when uploading the supplementary material to OpenReview, the version of supplementary material that shows up is still the old version. We reached out to the PCs and are waiting for them to help us update it. Meanwhile, please refer to the following anonymous [GitHub page](https://github.com/iclr-2023-paper-5926/iclr2023-paper-5926) to access the supplementary material: https://github.com/iclr-2023-paper-5926/iclr2023-paper-5926
>
> ### Q2: Would $w_i$ impact performance if used in the loss function?
> A2: We tried optimizing the weighted objective, but found that computing $w_i$ becomes numerically unstable when the dimension is high. For example, in case of $\ell_2$ loss, computing $w_i=\int_{\delta\tau_{i}}^{\tau_{i}}(2t)^{d}\frac{\Gamma(1+1/p)^{d}}{\Gamma(1+d/p)}dt=\frac{(2(1-\delta)\tau_{i})^{d+1}}{2(d+1)}\cdot\frac{\Gamma(1+1/p)^{d}}{\Gamma(1+d/p)}$ becomes unstable because $\Gamma(1+d/p)$ overflows.
>
> ### Q3: Why can Eqn.4 drop $\tau_i$?
> A3: Lemmas 2 and 3 assume fixed $\tau_i$’s - they cannot be changed when optimizing \theta because it would amount to changing the size of the neighbourhood over which probability density is integrated. If this were done, the objective values for two different \theta’s would become incomparable, because they would correspond to the probabilities of different events. Consequently, we only set $\tau_i$’s before solving each optimization problem in the curriculum and keep them fixed when solving the optimization problem itself. For details on this in the training procedure, see line 4 of Algorithm 1.
>
> ### Q4: Derivation in Lemma 2 appears for a fixed sample $z_j$
> A4: $z_j$ just represents a random variable following the standard normal distribution and is not specific to a particular $j$, since the distribution of all $z_j$’s are identical To improve clarity, we replaced $z_j$ with a random variable “z” in the manuscript.
> ### Q5: One would require continuity of  $p_\theta$ at the datapoints to assert the limit in the third to last line in Lemma 4’s proof exist.
> A5: Thanks for pointing it out, we’ve updated the manuscript to include the continuity condition.
> ### Q6: Low-dimensional examples
> A6: Thanks for the suggestion. We will provide a more intuitive low-dimensional example in the camera-ready.
> ### Q7: Different metrics such as LPIPS
> A7:  Indeed we tried the LPIPS metric and found it to work well. As a result, we used a weighted combination of LPIPS distance and $l_2$ distance as the distance metric $d(\cdot,\cdot)$ in our experiments. Empirically, we found that this produced the best results.
> ### Q8: Missing reference on page 5
> A8: Good catch, we have fixed it in the manuscript.

---

> > ### Comment · Reviewer_Me9N · 2022-11-23
> > **Response**
> >
> > Thank you to the authors for their response. I think that my main concerns have been addressed, namely the equivalence between optimizing the loss with and without weights $w_i$. I still think it would be great to include some low-dimensional examples to gain some geometric intuition of the approach. I will keep my positive score.

---

> > > ### Author Response · Authors · 2022-11-25
> > > **Thank you for the response**
> > >
> > > Thank you for the feedback and the suggestions, we will add more illustrative examples to better convey the intuitions behind our method in the camera-ready version.

---

### Author Response · Authors · 2022-11-20
**General Response**

We thank all reviewers for your time, constructive comments and unanimous appreciation of the proposed method and the good results. In particular, the reviewers remarked, "this paper presents a nice extension of previous work”(R Me9N), "Clear paper writing [...] consistent performance gain and promising results" (R jG1N), "the paper shows solid experimental results, both quantitatively and qualitatively" (R owTn), “A systematic level up of vanilla IMLE algorithm and obvious improvement of results" (R xM5B), “achieves stable and better results [...] a beautiful work” (R uhYC).

We found the comments and questions very helpful, which add value to our work. Here, we provide a one-sentence summary of our response to selected questions raised by the reviewers — please refer to our individual responses to each review for full details.
### Q1: Why $w_i$ can be ignored in the final objective function in Eqn. 5? (R jG1N)
A1: We have added a proof in the supplementary material that shows the equivalence of optimizing the unweighted objective (without $w_i$) and the weighted one (with $w_i$).
### Q2: $w_i$ may be impractical to implement for certain distance metrics. (R Me9N)
A2: $w_i$ does not need to be computed explicitly due to equivalence between weighted and unweighted objectives.
### Q3: Why consider this problem setting (few-shot image synthesis without auxiliary datasets)? (R owTn)
A3: This setting was proposed in prior works [a, b] and is important when auxiliary datasets are unavailable.
### Q4: Is precision a good metric in this setting and why does vanilla IMLE achieve slightly better precision? (R jG1N, owTn)
A4: In the few-shot setting, precision from [c] is not very meaningful (we explain why below); note that our method is significantly better on FID and recall and only very slightly (0.1%-0.6%) worse on precision.
### Q5: Is the proposed method limited to learning 100 faces of one person instead of learning 100 faces of different people? (R xM5B)
A5: In fact, the FFHQ subset consists of faces of different people, and the results show that our method does well on it.
### Q6: Why does the proposed objective achieve better results than vanilla IMLE? (R uhYC)
A6: The proposed Adaptive IMLE objective can better adapt to data points with varying difficulties compared to vanilla IMLE.
### Q7: Do you need additional large-scale data for pre-training? (R jG1N)
A7: No, our method can train from scratch.


### References
[a] Bingchen Liu, Yizhe Zhu, Kunpeng Song, and Ahmed Elgammal. Towards faster and stabilized GAN training for high-fidelity few-shot image synthesis.

[b] Chaerin Kong, Jeesoo Kim, Donghoon Han, and Nojun Kwak. Smoothing the generative latent space with mixup-based distance learning. In European Conference on Computer Vision, 2022.

[c] Tuomas Kynkäänniemi, Tero Karras, Samuli Laine, Jaakko Lehtinen, and Timo Aila. Improved precision and recall metric for assessing generative models. Advances in Neural Information Processing Systems, 32, 2019.

---

### Decision · Program_Chairs · 2023-01-20

**Decision:**

Reject

**Justification For Why Not Higher Score:**

The paper has a correctness issue.

**Justification For Why Not Lower Score:**

N/A

**Metareview: Summary, Strengths And Weaknesses:**

The paper is proposing to extend IMLE with weights and parameters depending on input data points. The method follows replacing min with a softer version to use interchangeability of expectation and summation later taking limiting behaviour by choosing a specific function class to make sure min and softer version agrees. The method is later tested on various different benchmarks and the results were very promising. The paper was reviewed by 5 experts and received borderline reviews. Later a reviewer AC calibraiton meeting was performed and the following major points are shared by reviewers:

- Positive aspects:
  - The application is very interesting and important
  - Results are good

- Negative aspects:
  - Novelty is limited
  - Scope is very narrow and experiments are also narrow
  - Technical correctness is not clear.

I believe the novelty issue is not a valid concern and the scope issue is solved by authors during rebuttal. However, technical correctness was remaining so I decided to read the paper in detail and checked all the proofs. I believe the paper has significant correctness issues which should have been clarified. Here are the major ones:

Before understanding the correctness issue, notice the following:
- In Lemma 1, $\Phi$ is a single function of $X$. Moreover, it is assumed to be integrable. Its proof is entirely correct and no correctness/clarity issue is there.
- In Lemma 2, The lemma is correct in its currently written form. However, there is a significant assumption hidden and not made explicit. This lemma is only correct if $\tau_i$ is independent of $z_k,x_k$ where $i \neq k$. If this is not true, probabilities and integrals are over the joint distributions not marginals. This is not made explicit anywhere in the paper, also not discussed.
- In Lemma 3, this is rather application of Lemma 2 so it has the same assumption.

In the rest of the paper, $\tau_i$ is chosen very carefully using nearest neighbours simply invalidating the original assumption. It depends on other points. Hence, lemmas can not be applied to the remaining derivation.

Another correctness issue is caught by the reviewers. Reviewers simply noticed that $w_i$ can not be simply ignored and they are right. Authors' counter argument is the fact that both optimization problems have the same optimal solution so they are equivalent. This is only true for convex optimization since having the same global optimum point does not mean anything about the equivalence of two non-convex optimization problems. Even for convex case, it simply means they are same asymptotically which does not imply the equivalence in finite sample case. Regardless, for finite regime and non-convex case, these two optimization problems are obviously not equivalent and authors can see it by simply setting $w_i=1$ for one point and setting $\frac{1}{w_i}=10^{-20}$ for all other points and running their method.

In summary, there are correctness issues with the paper. I believe there is a major correctness issue. Moreover, authors did not put any effort to make checking correctness easy. There are no motivation for correctness of any lemma or theorem is provided. Statements are very technical without accompanying discussion.

**Summary Of Ac-Reviewer Meeting:**

The AC reviewer meeting revolved around the following points:
- Positive points for the paper are raised as follows:
  - The application is very interesting and important
  - Results are good
  - Method is theoretically sound
- Negative points:
  - Novelty is limited (this was not unanimous and as an AC I disagreed)
  - Scope is very narrow and experiments are also narrow (this was mostly agreed but partially addressed using authors' additional experiments)
  - Technical correctness was not clear. Especially fact that w_i can be removed from the optimization and how it is removed was not clear. This resulted in questioning the technical correctness of the paper.

Our final agreement was the following:
- As an AC, I will check the theoretical correctness of the paper and will clarify the concerns of the reviewers.
- If the correctness issue is resolved, the decision will be based on applicability of the method and the generality of the scope.

---

> ### Author Response · Authors · 2023-02-12
> **Meta Review Response**
>
> Thank you for the thorough review and the recognition of the novelty of our method. We have carefully read the review and tried to incorporate useful suggestions to improve our paper. However, we disagree with the reasoning presented regarding correctness issues, as we believe it to be inaccurate. Below we provide more details.
> ## Choice of $\tau_i$
> In our algorithm, $\tau_i$ can be chosen independently of $z$ and $x$. In Algorithm 1, we initialized $\tau_i$ to be the closest sample distance to the target image but it is decreased in line $11$ of the Algorithm based on a fixed schedule. In fact, any large enough value (greater or equal to that closest distance) would work too for the initialization. Nevertheless, we will make it clear in future drafts to avoid confusion.
>
> ## Whether $w_i$ can be ignored in the objective function
> The meta-review questioned our proof of equivalence of the weighted and the unweighted optimization problems, in which we showed the optimal solutions to the two problems are equivalent. Specifically, the review said “This is only true for convex optimization since having the same global optimum point does not mean anything about the equivalence of two non-convex optimization problems. Even for convex case, it simply means they are same asymptotically which does not imply the equivalence in finite sample case." However, there is an important distinction between the equivalence of two optimization problems and the equivalence of the solutions found by gradient descent applied to the two problems. For example, maximizing log-likelihood is considered to be equivalent to maximizing likelihood. However, solutions found by gradient descent found for these two problems may not be the same. This is because the loss landscapes are different for these two problems so the convergence behaviour is also different. However, this does not mean that maximizing likelihood is not equivalent to maximizing log likelihood regardless of the convexity or the lack thereof. Convexity is only needed to guarantee convergence of gradient-based optimization to the global optimum, but as mentioned above, equivalence of two optimization problems is a property of the optimization problems themselves and is not specific to the choice of optimization algorithms.
>
> The convergence of optimization algorithms is of course dependent on the conditioning of the optimization problem, and two optimization problems can be equivalent and yet have different conditionings. An example would be minimizing Euclidean distance vs. square root of Euclidean distance - the former is better conditioned than the latter, but both have the same optimal solution. All the suggested experiment shows is that the weighted and unweighted optimization problems are conditioned differently, but this does not somehow disprove the equivalence of the optimization problems.